# Comparison of Central Aortic Pressure between Women with Preeclampsia and Normotensive Postpartum Women from an Urban Region of Western Mexico

**DOI:** 10.3390/medicina59071343

**Published:** 2023-07-21

**Authors:** Francisco J. Hernández-Mora, Claudia K. Cerda-Guerrero, Leonel García-Benavides, Enrique Cervantes-Pérez, Sol Ramírez-Ochoa, Janet Cristina Vázquez-Beltrán, Gabino Cervantes-Guevara, Ernesto Ledezma-Hurtado, Adriana Nápoles-Echauri, Alejandro González-Ojeda, Clotilde Fuentes-Orozco, María Isabel Hernández-Rivas, Mariana Chávez-Tostado, Guillermo A. Cervantes-Cardona

**Affiliations:** 1Department of Human Reproduction, Growth and Child Development, Centro Universitario de Ciencias de la Salud, Universidad de Guadalajara, 44340 Guadalajara, Jalisco, Mexico; frank.gine@gmail.com (F.J.H.-M.); ln.marianachavez@gmail.com (M.C.-T.); 2Department of Obstetrics, Hospital Civil de Guadalajara “Fray Antonio Alcalde”, Centro Universitario de Ciencias de la Salud, Universidad de Guadalajara, 44200 Guadalajara, Jalisco, Mexico; letyelindependiente@hotmail.com (C.K.C.-G.); ginapublicidad@gmail.com (E.L.-H.); 3Department of Biomedical Sciences, Centro Universitario de Tonalá, Universidad de Guadalajara, 45425 Tonalá, Jalisco, Mexico; cginde02@yahoo.com.mx; 4Department of Internal Medicine, Hospital Civil de Guadalajara “Fray Antonio Alcalde”, Centro Universitario de Ciencias de la Salud, Universidad de Guadalajara, 44200 Guadalajara, Jalisco, Mexico; enrique19896@msn.com (E.C.-P.); ramirez_ochoa_sol@hotmail.com (S.R.-O.); 5Department of Clinics, Centro Universitario de Tlajomulco, Universidad de Guadalajara, 45641 Tlajomulco de Zúñiga, Jalisco, Mexico; 6School of Medicine, Instituto Politécnico Nacional, 11340 Ciudad de México, Ciudad de Mexico, Mexico; janet.cris.beltran@gmail.com; 7Department of Welfare and Sustainable Development, Centro Universitario del Norte, Universidad de Guadalajara, 46200 Colotlán, Jalisco, Mexico; gabino_guevara@hotmail.com; 8Department of Philosophical, Methodological and Instrumental Disciplines, Centro Universitario de Ciencias de la Salud, Universidad de Guadalajara, 44340 Guadalajara, Jalisco, Mexico; izardetheodoroth@gmail.com; 9Biomedical Research Unit 02, Hospital de Especialidades, Centro Médico Nacional de Occidente, 44329 Guadalajara, Jalisco, Mexico; avygail5@gmail.com (A.G.-O.); clotilde.fuentes@gmail.com (C.F.-O.); 10Odontology Department for the Preservation of Health, Centro Universitario de Ciencias de la Salud, Universidad de Guadalajara, 44340 Guadalajara, Jalisco, Mexico; misabel.hernandez@academicos.udg.mx

**Keywords:** central aortic pressure, preeclampsia, puerperium, pregnancy, augmentation index

## Abstract

*Background and Objectives*: Central aortic pressure (CAP) can be measured through noninvasive methods, and CAP wave analysis can provide information about arterial stiffness. The objective of this study was to compare CAP in women with preeclampsia and normotensive postpartum women from an urban region in western Mexico. *Materials and Methods:* We recruited 78 women in immediate puerperium, including 39 with preeclampsia and 39 with normotension, who received delivery care in our hospital between September 2017 and January 2018. Pulse wave analysis was used to assess central hemodynamics as well as arterial stiffness with an oscillometric device. For this purpose, the measurement of the wave of the left radial artery was obtained with a wrist applanation tonometer and the ascending aortic pressure wave was generated using the accompanying software (V 1.1, Omron, Japan). Additionally, the systolic CAP, diastolic pressure, pulse pressure, heart rate, and rise rate adjusted for a heart rate of 75 bpm were determined. The radial pulse wave was calibrated using the diastolic and mean arterial pressures obtained from the left brachial artery. For all the statistical analyses, we considered *p* < 0.05 to be significant. *Results:* The results were as follows: a systolic CAP of 125.40 (SD 15.46) vs. 112.10 (SD 10.12) with *p* < 0.0001 for women with and without preeclampsia, respectively. Systolic CAP was significantly elevated in women with preeclampsia and could indicate an elevated risk of cardiovascular disease. *Conclusion:* CAP is an important parameter that can be measured in this group of patients and is significantly elevated in women with postpartum preeclampsia, even when the brachial blood pressure is normal.

## 1. Introduction

There is currently a range of conditions that include pregnancy-related hypertensive disorders, which are classified before and after the 20th week of gestation [1]. Chronic hypertension is classified before the 20th week of gestation, while gestational hypertension, preeclampsia/eclampsia, and chronic hypertension with superimposed preeclampsia occur from this period onward [1].

Hypertensive disorders in pregnancy are the second leading cause of maternal death, accounting for 14.0% of maternal deaths around the world, and they are an important cause of morbidity in this group of patients [2]. Regarding the specific epidemiology of Mexico, by 2022, 30.4 maternal deaths were reported for every 100,000 births; hypertensive disease, edema, and proteinuria during pregnancy, childbirth, and puerperium constituted the second leading cause of death at 17.2% [3] (DGE, 2022).

Preeclampsia (PE) is defined as a disorder of pregnancy associated with new-onset arterial hypertension after the 20th week of gestation until the first 6 weeks postpartum, commonly accompanied by recent-onset proteinuria; however, this condition may present as other signs or symptoms in the absence of proteinuria [4]. The main risk factors are multiple gestations, maternal age over 35 years, the use of assisted reproductive technology, and preeclampsia in previous pregnancies, among others [4].

The diagnosis of PE is complex since it implies the involvement of different systems and not only the existence of arterial hypertension (>140/90 mmHg); for example, patients with PE can present with proteinuria (2+ on a test strip or >300 mg from a 24 h urine sample) and/or evidence of acute kidney injury, liver dysfunction, thrombocytopenia, hemolysis, neurological features, and fetal growth restriction [1,5]. Symptoms that patients may present with include headache, visual disturbances, epigastric or right upper quadrant pain, and edema [1,5].

Cardiovascular function during pregnancy is affected by important changes that allow for healthy fetal development, for example, a 20% increase in pulse rate, uterine artery blood flow, a 30–40% increase in cardiac output, blood flow microvascular, and blood flow at rest; likewise, blood pressure decreases at approximately 18 weeks gestation and then increases toward the end of pregnancy [1]. This stress of high pulse pressure subsequently leads to arterial stiffness through fissure due to exhaustion of the elastic components of the arterial wall [6].

In recent years, the association of PE with increased arterial stiffness during and after pregnancy, rather than gestational hypertension, has been documented [7,8,9]. On the other hand, a considerable increase in arterial stiffness has been seen in patients with underlying conditions, such as gestational diabetes and PE, who showed greater alterations in the arterial wall, followed by patients with hypertension, compared to pregnant patients with normotension [7,8,9]. In addition, patients with a history of or current PE have an increased cardiovascular risk due to this endothelial dysfunction [7,8,9].

The evaluation of arterial function can be performed through invasive and noninvasive methods, where the latter are preferred since they are safe, easy, and reproducible procedures [1,6,8,10,11]. Vascular health assessment is performed through peripheral arterial tonometry using central aortic pressure (CAP) and the augmentation index (AIx) as indirect parameters to measure the pulse wave velocity throughout the arterial tree, which is inversely related to elasticity and whose values are increased in patients with preeclampsia [1,6,8,10,11].

Specifically, the AIx is both an indicator of left ventricular work during systole and a direct measure of vasoconstriction, which is why it has been shown to be an independent predictor of cardiovascular risk, coronary disease, cerebrovascular events, cardiovascular mortality, and exercise tolerance [10,12,13,14]. On the other hand, CAP can be used as an independent predictor of cardiovascular risk and cardiovascular mortality and is a marker of ventricular load [10,12,13,14]. It is also useful in the assessment of hypertensive states, pharmacological treatments, and in the evaluation of target organ damage [12,13,14]. These parameters contribute to the early diagnosis and identification of patients suffering from some type of hypertension and those at risk of developing hypertension, as well as their cardiovascular risk [10,12,13,14].

Given the high rates of PE in pregnant Mexican women and the fact that there are no studies that report its association with arterial stiffness, this descriptive study was proposed, with the aim of comparing CAP between women with preeclampsia and women with normotension from an urban region in western Mexico.

## 2. Materials and Methods

This study was performed in the obstetrics department of the Fray Antonio Alcalde Civil Hospital from 1 September 2017 to 19 January 2018.

For this study, a sample size of 78 subjects, 39 in each arm, is sufficient to detect a clinically important difference of 7.3 between groups, assuming a standard deviation of 11.25 using a two-tailed t-test of the difference between means with 80% power and a 5% level of significance (39 patients without preeclampsia and 39 patients with preeclampsia).

The recruitment criteria for this study were as follows: women in immediate puerperium who were willing to participate in the study and who gave informed consent before any protocol procedures. All participants had no history of any other cardiovascular disease except PE at the time of inclusion.

PE was defined as follows: a systolic blood pressure of 140 mmHg or more and diastolic blood pressure of 90 mmHg or more on two independent measurements taken at least 4 h apart in combination with proteinuria (≥300 mg/dl total protein in a 24 h urine collection and, if this was not available, ≥ +2 proteinuria on a dipstick) in pregnant women who were previously normotensive for at least the first 20 weeks of pregnancy to through the first 6 weeks postpartum, according to the PE diagnostic criteria of the American College of Obstetricians and Gynecologists [4]. After completing the medical records, patients were instructed to remain at rest for 15 min to establish stable hemodynamic conditions. Peripheral systolic and diastolic blood pressure was measured in the right arm.

Pulse wave analysis was used to assess central hemodynamics as well as arterial stiffness with the oscillometric device Omron HEM-9000 AI (Omron, Japan). For this purpose, the measurement of the wave of the left radial artery was obtained with a wrist applanation tonometer and an ascending aortic pressure wave was generated using the accompanying software (V 1.1, Omron, Japan). Additionally, the systolic central aortic pressure, diastolic pressure, pulse pressure, heart rate, and rise rate adjusted for a heart rate of 75 bpm were determined. The radial pulse wave was calibrated using the diastolic and mean arterial pressures obtained from the left brachial artery. All these procedures were performed in accordance with the instructions provided by the manufacturer [14,15].

The study was conducted in accordance with the Declaration of Helsinki (as revised in 2013). The Clinical Research and Bioethics Committee of the Hospital Civil de Guadalajara Fray Antonio Alcalde approved this study, and all patients signed informed consent forms.

### Statistical Analysis

All data obtained were analyzed using GraphPad Prism version 9.3.1.471 (GraphPad by Dotmatics, America). For the descriptive analysis, quantitative variables were analyzed with measures of central tendency and dispersion (the median and standard deviation). For inferential analysis, the differences between groups were analyzed using Student’s t test, and for nonparametric data, the Mann-Whitney U test was used. For all statistical analyses, we considered *p* < 0.05 to be significant.

## 3. Results

We recruited 78 women in immediate puerperium for our study, 39 patients with preeclampsia and 39 patients with normotension, who received delivery care in our hospital between September 2017 and January 2018.

Of the included patients diagnosed with preeclampsia, 6 were diagnosed with early-onset preeclampsia (diagnosed before 34 weeks’ gestation), while 33 were diagnosed with late-onset preeclampsia (diagnosed after 34 weeks’ gestation). Patients were classified as having severe preeclampsia if, at the time of diagnosis, they presented pressures above 160/110 mmHg, proteinuria, and headaches accompanied by phosphenes.

At the time of inclusion in this study, all the patients with PE (n = 39) were under antihypertensive treatment, of which 14 patients (35.89%) were treated with captopril and nifedipine; 12 patients (30.76%) with captopril alone; 8 patients (20.52%) with enalapril and nifedipine; and 5 patients (12.83%) with losartan and nifedipine. No other medications (both for women with and without preeclampsia) were taken during the collection of data for our study.

For our sample of patients, the following clinical characteristics were analyzed, comparing the group of women with preeclampsia with the group of women without preeclampsia: age, with a mean of 25.59 (standard deviation, SD ± 7.5) vs. 22.9 (SD ± 5.7) years, with a value of *p* = 0.882 and a confidence interval (CI) of 0–5; weight, with a mean of 74.01 (SD ± 18.03, CI) vs. 70.09 (SD ± 15.71) kilograms, with a value of *p* = 0.3088 and a CI of −3.70–11.55; body mass index (BMI) of 28.73 (SD ± 6.23) vs. 28.25 (SD ± 5.37) kg/m^2^ with a value of *p* = 0.7154 and a CI of −2.14–3.1; gestational age at delivery of 36.68 (SD ± 2.83) vs. 39.21 (SD ± 1.34) weeks, with a value of *p* < 0.0001 and a CI of −3–−1; serum hemoglobin of 12.77 (SD ± 1.50) vs. 12.70 (SD ± 1.75) g/dl with a value of *p* = 0.864 and with a CI of −0.67–0.80, respectively. In patients with preeclampsia, the mean urine protein collected per 24 h was 1076.12 mg/24 h (SD ± 1703.63).

The comparison of the hemodynamic characteristics between women with and without preeclampsia can be seen in Table 1.

## 4. Discussion

In this study, we found that preeclampsia was characterized by an increase in systolic CAP; however, this increase was not observed in the AIx or in the AIx adjusted for a heart rate of 75 bpm in patients with preeclampsia compared to those without preeclampsia. In Austria in 2013, Franz et al. [12] reported a significant difference (*p* < 0.001) in the comparison of both groups for the AIx, while Turi et al. showed an equal result in pregnant Romanian women in 2020 [9].

On the other hand, regarding the AIx adjusted for a heart rate of 75 bpm, Torrado et al. showed a relevant significant difference between the evaluated Uruguayan groups (*p* = 0.05) [16], and in 2011, Savvidou et al. noted that the rate of increase in the AIx adjusted for a heart rate of 75 bpm was similar between their two groups of patients (*p* = 0.44) [17]. According to Namugowa et al.’ 2017 work, statistical significance was shown (*p* = 0.000) for both the AIx and the AIx adjusted for a heart rate of 75 bpm in the studied pregnant South African women [18]. In our study, the AIx did not show a statistically significant difference between the two groups, which may be related to the use of drugs that modify arterial distensibility, since calcium channel blockers (nifedipine) and/or angiotensin-converting enzyme inhibitors (captopril) were administered to all patients with preeclampsia in our sample [16]. Although circulating levels of angiotensin II are not increased in patients with preeclampsia, the vascular response to angiotensin II is increased [16,19]. Even though the AIx is a good independent indicator of vascular risk, its elevation is related to changes in left ventricular contractility, wave reflexes, and left ventricular preload [20]. In the case of our patients, the diagnosis of the hypertensive pathology is recent (so they still do not present ventricular changes or chronic alterations) and they are being treated with antihypertensive drugs (as mentioned above) that help to reduce the burden of the left ventricle.

Our study showed that the patients with preeclampsia in our sample presented a significant increase in central pressures, which suggests an increase in arterial stiffness in patients affected by preeclampsia compared to those without preeclampsia. Similarly, Torrado et al. found a significant elevation of this parameter in patients with preeclampsia [16]. Polónia J et al. conducted a study in 2014 [21] consisting of the evaluation of the central hemodynamic properties of the arterial wall in women with a history of preeclampsia, and an increase in CAP and peripheral vascular resistance, which contributes to a greater cardiovascular risk after pregnancy, was found in these women [1].

Regarding somatometry, our results for weight and BMI did not show statistical significance, in agreement with what was reported by Franz et al. in 2013 and by Namugowa et al. (weight, *p* = 0.115 and BMI, *p* = 0.067) [12,18]. This was in contrast to what was reported by Savvidou et al., Turi et al., and Torrado et al. (*p* < 0.001 for both variables in the three studies) [9,16,17].

In our study, we observed a statistically significant difference in terms of gestational age at delivery, being lower in the case of women with preeclampsia. This can be explained due to the recommendations of international clinical guidelines regarding indications for induction of labor or caesarean section in women with preeclampsia, taking into account that our sample of patients was only women with severe preeclampsia [22].

The heart rate in our study was not different between the groups examined, which was also reported by Savvidou et al. in 2011 (*p* = 0.22) and Torrado in 2015 (*p* = 0.844) [16,17]. Conversely, in the work of Turi et al. from 2020 (*p* = 0.006) and Namugowa et al. from 2017 (*p* = 0.000), significant differences in this parameter were not found [9,18].

Pulse pressure did not show a difference between the groups in our study, similar to that reported by Savvidou et al. in UK patients (*p* = 0.36) [17]. In contrast, both Torrado et al. (*p* = 0.041) and Namugowa et al. (*p* = 0.000) obtained statistical relevance in their respective populations [16,18].

In our study, pregnant patients with preeclampsia had higher systolic, diastolic, and mean arterial pressure (MAP) values than pregnant women with normotension. These results are similar to those found by Namugowa et al. in 2017 (*p* = 0.000, for the three pressures), and although the averages of these variables reported in the publication were below the diagnostic blood pressures for hypertension, the rest of the endothelial function evaluations were altered [18]. However, this phenomenon was not presented by pregnant women with preeclampsia in the study by Torrado et al. (*p* < 0.001 for the three pressures), Franz et al., or Turi et al. (*p* < 0.001 in both studies), predominantly in systolic and diastolic pressures [9,12,16].

In patients with preeclampsia, the measurement of CAP showed that despite normal BBP values (even when these patients presented statistically higher systolic and diastolic blood pressure values), CAP remained high [23]; therefore, this evaluation provides us with important data on arterial stiffness, which is closely related to cardiovascular disease, heart failure, stroke, acute myocardial infarction, hypertensive crisis, and eclampsia [1,9,24,25,26].

CAP can be measured with a noninvasive method that provides relevant data on short- and long-term therapeutic behaviors [15]. It should be noted that there is no guide worldwide that indicates the precise moment to suspend antihypertensive therapy in women with preeclampsia; thus, radial artery applanation tonometry could provide guidelines for the long-term follow-up of these patients.

One limitation of our study is that as it was a cross-sectional study, we cannot say whether the increase in systolic CAP in patients with preeclampsia was the cause of the disease or a consequence of the pathology.

## 5. Conclusions

Even though the sample size of our study was relatively small, the means and standard deviations in our study are similar to those of other studies that have statistical significance, indicating that the sample of 78 patients was adequate to detect the effect of pregnancy status with at least 80% power.

The measurement of systolic CAP in patients with preeclampsia who are given a subsequent clinical follow-up could be a more reliable indicator of the cardiovascular health of the patients and indicate the need for targeted interventions to avoid chronic cardiovascular changes or reduce the mortality associated with the lack of periodic medical check-ups in patients with high cardiovascular risk.

In conclusion, women with preeclampsia in our study presented significantly higher systolic pressure, diastolic pressure, mean arterial pressure, and systolic CAP than women without preeclampsia. CAP is an important parameter that can be measured in this group of patients and is significantly elevated in women with preeclampsia, even when the BBP is normal. It can indicate an elevated risk of cardiovascular disease that warrants follow-up for possible therapeutic interventions, and its noninvasive measurement can be useful in a variety of clinical situations, such as early recognition of PE. Future longitudinal studies in this group of patients are needed to determine causality.

## Figures and Tables

**Table 1 medicina-59-01343-t001:** Comparison of the hemodynamic characteristics between women with and without preeclampsia.

Characteristic	With PE, n = 39 (Mean ± SD)	Without PE, n = 39 (Mean ± SD)	95% IC	*p* Value
Systolic pressure (mmHg)	124.1 ± 15.48	111 ± 7.61	7.53–18.53	<0.0001 *
Diastolic pressure (mmHg)	78.92 ± 13.04	67.82 ± 8.23	6.18–16.02	<0.0001 *
Mean arterial pressure (mmHg)	93.26 ± 13.48	81.59 ± 7.77	6.70–16.63	<0.0001 *
Augmentation index (%)	70.15 ± 11.25	72.77 ± 13.85	−8.30–3.07	0.3630
AIxP75 (%)	77.90 ± 10.68	80.10 ±12.95	−7.55–3.14	0.4146
Central aortic pressure (mmHg)	125.4 ± 15.46	112.10 ± 10.12	7–18	<0.0001 *
Pulse pressure (mmHg)	45.15 ± 7.96	43.41 ± 4.97	−1.25–4.73	0.2497
Heart rate (bpm)	93.23 ± 15.38	92.67 ± 13.08	−5.87–7	0.8619

SD = standard deviation, PE = preeclampsia, AIxP75 = augmentation index adjusted to 75 beats per minute (bpm), CI = confidence interval, * *p* ≤ 0.05.

## Data Availability

The data presented in this study are available on request from the corresponding author. The data are not publicly available due to privacy reasons.

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
