# Peer review of "Comparison of Central Aortic Pressure between Women with Preeclampsia and Normotensive Postpartum Women from an Urban Region of Western Mexico"

_medicina, 2023, doi:10.3390/medicina59071343_

Round 1
Reviewer 1 Report
The manuscript is original and well defined. Regarding the structure and accuracy of the phrases, the artical has well structured information, with supported evidence and well structured phrases.
The results provide an advance in current knowledge. The results are being interpreted appropriately and are significant, as well as the conclusions.
Surely the paper will attract a wide readership.
Author Response
We appreciate your valuable comments, they are motivating to continue with our future research.
Reviewer 2 Report
Dear Authors,
The manuscript is generally well-written; however, there are some points to be improved:
The description of the results is too short. Although there is no statistical difference in clinical characteristics between patients with PE and normotensive women, in my opinion, it would be better to give the description separately in the text, and not the average values for all the women under study
Additional data (if possible) for the clinical status of the women should be provided in Table 1 such as proteinuria, liver function tests, gestational age at delivery, etc.
What type of preeclampsia did you study - severe or mild type; early onset type or late-onset type?
It should be given information if the patients have taken some medications.
Author Response
Dear Authors,
The manuscript is generally well-written; however, there are some points to be improved:
1. The description of the results is too short. Although there is no statistical difference in clinical characteristics between patients with PE and normotensive women, in my opinion, it would be better to give the description separately in the text, and not the average values for all the women under study.
According to your suggestion, in order not to be redundant, we delete Table 1, add some additional results (which were also suggested to us later) and describe them in text form, as well as remove the means for the global sample which we agree, appear redundant when evaluating the values for each group.
2. Additional data (if possible) for the clinical status of the women should be provided in Table 1 such as proteinuria, liver function tests, gestational age at delivery, etc.
Unfortunately, of the requested data, we only have the weeks of gestation; this comparison was added to the results, and discussed briefly in the discussion section. In addition to this, we added the hemoglobin values for both groups and the mean urine protein collected in 24 hours for the patients with preeclampsia (this is because the group without preeclampsia did not have proteinuria).
3. What type of preeclampsia did you study - severe or mild type; early onset type or late-onset type?
The patients included in our study were classified as having severe preeclampsia, 6 with early onset and 33 with late onset. A more detailed description is added in the results section.
4. It should be given information if the patients have taken some medications.
We included the medications taken by our patients in the results section
Reviewer 3 Report
1. The objective of this study was to compare the CAP of postpartum women with preeclampsia and the normal control. However, It seems to have little clinical useful. Right now, there's no further intervention or prevention of deterioration of cardiac function in women with history of preeclampsia.
2. The study was well-designed. Anyway, please give more information regarding the sample size calculation.
3. Please discuss more why AI of PE and non-PE women was not different.
4. Please discuss the clinical applicability of this study.
Author Response
- The objective of this study was to compare the CAP of postpartum women with preeclampsia and the normal control. However, It seems to have little clinical useful. Right now, there's no further intervention or prevention of deterioration of cardiac function in women with history of preeclampsia.
For this reason, we believe that generating information regarding the clinical characteristics of this group of patients could give us a better idea of the cardiovascular status of the patients allow us to use these as possible future therapeutic targets.
- The study was well-designed. Anyway, please give more information regarding the sample size calculation.
We added a better description of the sample size in the methods section.
- Please discuss more why AI of PE and non-PE women was not different.
Some reasons that could explain this finding were added in the discussion section.
- Please discuss the clinical applicability of this study.
Added a little more to the discussion section regarding clinical applicability.